# Ahf-Caltide, a Novel Polypeptide Derived from Calpastatin, Protects against Oxidative Stress Injury by Stabilizing the Expression of Ca_V_1.2 Calcium Channel

**DOI:** 10.3390/ijms242115729

**Published:** 2023-10-29

**Authors:** Yingchun Xue, Shi Zhou, Ling Yan, Yuelin Li, Xingrong Xu, Xianghui Wang, Etsuko Minobe, Masaki Kameyama, Liying Hao, Huiyuan Hu

**Affiliations:** 1Department of Pharmaceutical Toxicology, School of Pharmacy, China Medical University, Shenyang 110122, China; yingchunxue@yeah.net (Y.X.); zhoushicmu@163.com (S.Z.); yanling1074@163.com (L.Y.); lyl184082247651996@126.com (Y.L.); xingrongxu2023@163.com (X.X.); wangxhui321@163.com (X.W.); 2Department of Physiology, Graduate School of Medical and Dental Sciences, Kagoshima University, Kagoshima 890-8544, Japan; mimiben@m3.kufm.kagoshima-u.ac.jp (E.M.); mkame009@yahoo.co.jp (M.K.)

**Keywords:** Ahf-caltide, myocardial oxidative stress injury, Ca_V_1.2 calcium channel, calpastatin domain L

## Abstract

Reperfusion after ischemia would cause massive myocardial injury, which leads to oxidative stress (OS). Calcium homeostasis imbalance plays an essential role in myocardial OS injury. Ca_V_1.2 calcium channel mediates calcium influx into cardiomyocytes, and its activity is modulated by a region of calpastatin (CAST) domain L, CS_L_54-64. In this study, the effect of Ahf-caltide, derived from CS_L_54-64, on myocardial OS injury was investigated. Ahf-caltide decreased the levels of LDH, MDA and ROS and increased heart rate, coronary flow, cell survival and SOD activity during OS. In addition, Ahf-caltide permeated into H9c2 cells and increased Ca_V_1.2, Ca_V_β2 and CAST levels by inhibiting protein degradation. At different Ca^2+^ concentrations (25 nM, 10 μM, 1 mM), the binding of CS_L_ to the IQ motif in the C terminus of the Ca_V_1.2 channel was increased in a H_2_O_2_ concentration-dependent manner. CS_L_54-64 was predicted to be responsible for the binding of CS_L_ to Ca_V_1.2. In conclusion, Ahf-caltide exerted a cardioprotective effect on myocardial OS injury by stabilizing Ca_V_1.2 protein expression. Our study, for the first time, proposed that restoring calcium homeostasis by targeting the Ca_V_1.2 calcium channel and its regulating factor CAST could be a novel treatment for myocardial OS injury.

## 1. Introduction

Acute myocardial infarction is the leading cause of death and disability worldwide. The essential remedy for ischemia is to restore blood flow timely. However, reperfusion after ischemia can cause additional cellular damage, termed as ischemia–reperfusion (IR) injury, which leads to cardiac dysfunction such as myocardial stunning, reperfusion arrhythmia and myocyte death [1,2,3]. Among the complex system networks involved in the pathological mechanisms of IR injury, oxidative stress (OS) is one of the most important pathological mechanisms [1]. OS refers to the imbalance between reactive oxygen species (ROS) production and cellular antioxidant capacity [4]. Elevated ROS induces the influx of extracellular Ca^2+^ via the voltage-sensitive Ca^2+^ channel opening, which is related with myocardial OS injury [1,5].

Ca_V_1.2 calcium channel is the main route for Ca^2+^ influx into cardiomyocytes, which triggers the release of intracellular Ca^2+^ for excitation–contraction coupling. The Ca_V_1.2 channel is a heterotetramer composed of α_1_, α_2_δ, β_2_ and γ subunits. The α_1_ subunit consists of four transmembrane regions, three intramembrane loop regions, N terminus (NT) and C terminus (CT) [6,7]. The dysregulation of intracellular Ca^2+^ homeostasis due to Ca_V_1.2 channels plays an important role in myocardial OS injury, and Ca_V_1.2 channel activity was altered after exposure to ROS [6,8]. Accordingly, the calcium homeostasis imbalance during OS may be related with myocardial injury. The alternations in the density or activity of Ca_V_1.2 channel and elevated ROS level have been reported to contribute to myocardial OS injury [1,8]. Therefore, restoring calcium homeostasis by regulating the Ca_V_1.2 channel may benefit in alleviating myocardial OS injury.

Calpastatin (CAST) is a calcium-dependent cysteine protease, composed of a unique N-terminal region (domain L, CS_L_) and four homologous C-terminal regions (domains 1–4) [9,10,11]. CS_L_ partially reactivates the Ca_V_1.2 channel, and this is mediated by the region containing amino acid (a.a.) residues 54-64 (EGKPKEHTEPK, CS_L_54-64) [9,12,13,14]. Subsequent studies have confirmed that CS_L_ binds to PreIQ and IQ motifs on the CT of the α_1_ subunit of the Ca_V_1.2 channel [15,16]. CAST localizes in the area of the Z-disk, around which Ca_V_1.2 channels in the T-tubular system are also localized [15]. Thus, it is reasonable to think that the concentration of CAST is high enough to affect the activity of the Ca_V_1.2 channel, which may be related with myocardial OS injury.

The α_1_ subunit of the Ca_V_1.2 channel is a therapeutic target in OS-induced myocardial injury [17,18,19,20,21]. Peptides and peptidomimetics have been widely used for the prevention, diagnosis and treatment of cardiovascular diseases [22,23]. In this study, with the target being the α_1_ subunit of the Ca_V_1.2 channel, Ahf-caltide, a polypeptide derived from CS_L_54-64, was investigated. Ahf-caltide significantly alleviated myocardial injury induced by OS, followed by the upregulation of the Ca_V_1.2 channel α_1_ subunit (Ca_V_1.2) and β2 subunit (Ca_V_β2) levels. In addition, with the increase in H_2_O_2_ concentration, the binding of CS_L_ to IQ motif was gradually increased and CS_L_54-64 was responsible for CS_L_ binding to Ca_V_1.2. Therefore, Ahf-caltide might be a promising therapeutic target in the prevention of myocardial OS injury.

## 2. Results

### 2.1. Protective Effect of Ahf-Caltide on H_2_O_2_-Induced Oxidative Stress Injury

In H_2_O_2_-induced OS injury in rats, compared with the control group, the heart rate and coronary flow were decreased (Figure 1B,C, *p* < 0.05). However, compared with the H_2_O_2_ group, the heart rate and coronary flow increased with the treatment of Ahf-caltide (*p* < 0.01, Figure 1B,C). In addition, LDH content significantly increased after H_2_O_2_ injury, which was decreased after Ahf-caltide treatment (Figure 1D). Furthermore, compared with the H_2_O_2_ group, BNP level was markedly decreased after Ahf-caltide treatment (Figure 1E,F).

The Ca_V_1.2 channel is significantly glutathionylated by ROS, which contributes to the development of cardiac pathology and is a therapeutic target in heart diseases [16,17,18,19,20]. CS_L_54-64 was reported to activate the Ca_V_1.2 channel [9]. Therefore, the protective mechanism of Ahf-caltide against H_2_O_2_-induced OS injury was studied from the perspective of the Ca_V_1.2 channel. The specificity of Ca_V_1.2 antibody was verified, which is shown in Appendix A. It was notable that the Ca_V_1.2 protein level was decreased to 51% of the control with H_2_O_2_ treatment, which was significantly increased by Ahf-caltide (Figure 1E,F). There was no significant difference in the Ca_V_β2 level among groups. The expression of CAST was higher in the H_2_O_2_ group or Ahf-caltide group than that in the control group. However, the CAST level showed no difference between the H_2_O_2_ group and Ahf-caltide group. The above results suggested that Ahf-caltide had a protective effect on H_2_O_2_-induced myocardial OS injury by up-regulating the Ca_V_1.2 level.

### 2.2. Protective Effects of Ahf-Caltide on Cardiac IR Injury

In the in vitro IR model, similar results were detected. Ahf-caltide relieved the changes in heart rate, coronary flow and LDH activity induced by IR injury in a dose-dependent manner (Figure 2B–D). SOD activity was increased and MDA level was decreased in the 10 μM Ahf-caltide group (Figure 2E,F). These results indicated that Ahf-caltide might have a protective effect on IR injury by enhancing the endogenous cellular anti-oxidant capacity. There was no significant difference in Ca_V_1.2 and Ca_V_β2 protein levels among groups (Figure 2G,H). CAST level decreased to 73.40 ± 8.67% (*n* = 5) of the control levels in IR injury hearts, while Ahf-caltide increased the level of CAST protein. These results suggested that the protective effect of Ahf-caltide might be mediated via up-regulation of the CAST level.

### 2.3. Ahf-Caltide Alleviated Oxidative Stress Injury in H9c2 Cells by Inhibiting Protein Degradation

Ahf-caltide conjugated with rodamine was used to examine the permeability of the peptide across the membrane and, if so, its cellular localization. As a result, immunofluorescence analysis identified that Ahf-caltide was localized in the cytoplasm of H9c2 cells evenly (Figure 3A). In addition, as shown in Figure 3B,C, the cell viability was significantly decreased in the CoCl_2_ group compared to the control group and was increased by Ahf-caltide at a concentration of 0.3, 1, 3 or 10 μM (*p* < 0.05). LDH activity was significantly increased to two- to three-fold following exposure to CoCl_2_ compared to that in the control group, which was markedly reduced in the H9c2 cells pretreated with Ahf-caltide (Figure 3B,C). Meanwhile, compared to the control group, the ROS level was significantly increased in the CoCl_2_ group, which decreased with the treatment of Ahf-caltide (Figure 3D,E).

The levels of Ca_V_1.2, Ca_V_β2 and CAST proteins were further explored. They were down-regulated in the CoCl_2_ group, which were reversed by Ahf-caltide (Figure 3F,G). To examine whether the changes in the expression of the Ca_V_1.2 channel were due to de novo synthesis, cycloheximide (CHX), an inhibitor of protein synthesis, was employed in the presence or absence of 10 μM Ahf-caltide for up to 36 h. Following the treatment of CHX, Ca_V_1.2, Ca_V_β2 and CAST levels showed a time-dependent decrease in normal cells. However, Ahf-caltide increased the protein levels of Ca_V_1.2, Ca_V_β2 and CAST in CHX-exposed cells (Figure 3H,I), suggesting that protein synthesis was not involved in the regulation of Ahf-caltide. It was reported that the ubiquitin–proteasome system and the autophagy–lysosomal pathway were involved in Ca_V_1.2 channel degradation [24]. Thus, the proteasome inhibitor MG-132 (10 µM) and lysosomal inhibitor leupeptin (10 µM) were applied. It was found that CAST, Ca_V_1.2 and Ca_V_β2 levels were increased by MG-132, which were further increased by Ahf-caltide. In contrast, there was no difference in Ca_V_1.2 and Ca_V_β2 levels between the leupeptin group and CoCl_2_ group (Figure 3J,K). However, CAST level was increased by leupeptin and then further increased under Ahf-caltide treatment (Figure 3J,K). These data strongly supported that proteasome inhibition was involved in the increase in CAST, Ca_V_1.2 and Ca_V_β2 mediated by Ahf-caltide, and lysosomal inhibition also was involved in the increase in CAST.

### 2.4. The Binding Characteristics of CS_L_ to Ca_V_1.2 Were Affected by Oxidative Stress

In our previous study, it was shown that CS_L_ could bind to the CT1 of Ca_V_1.2 [15,16,25]. But it was not known whether the binding characteristics changed during OS. In the present study, GST pull-down assay was used to examine the effect of OS on the binding of CS_L_ to Ca_V_1.2.

As was shown in Figure 4A,B, 10 μΜ CS_L_ with various concentrations of H_2_O_2_ (0 to 10^-1^ M) treatment was incubated with CT1 at [Ca^2+^] of 25 nM, 10 μΜ and 1 mM. The binding densities of CS_L_ to CT1 were increased with increasing concentrations of H_2_O_2_ (0 to10^−2^ M) at any [Ca^2+^] and increased with increasing [Ca^2+^] at fixed H_2_O_2_ concentration including the H_2_O_2_-free condition. Notably, at [Ca^2+^] of 25 nM, 10 μM and 1 mM, compared with the 10^−2^ M H_2_O_2_ group, the CS_L_ binding to CT1 was decreased by 10^−1^ M H_2_O_2_. Moreover, the CS_L_ bound to CT1 in the SDS–PAGE gel appeared in double bands with 10^−2^ M H_2_O_2_ treatment and was shifted up in the 10^−1^ M H_2_O_2_ group related to the treatment of H_2_O_2_ (0 to10^−2^ M). Then, the effects of H_2_O_2_ on the CS_L_ binding to PreIQ or IQ were examined. In 10 μM [Ca^2+^], the amount of CS_L_ binding to PreIQ was not significantly different compared to the control (Figure 4C,D, Table 1). However, the effect of H_2_O_2_ on the CS_L_ binding to IQ was similar to that to CT1 (Figure 4E,F, Table 1).

NT contained calmodulin (CaM) binding sites, which was reported to be involved in the regulation of Ca_V_1.2 channel activity [26]. But it is still unknown whether CS_L_ could bind to NT. Thus, the interaction of CS_L_ with NT was examined using pull-down assay. Our results showed that CS_L_ bound to NT in a [CS_L_]-dependent manner (Figure 4G,H). However, the concentration of Ca^2+^ has no effect on the binding of CS_L_ to GST-NT (Figure 4I,J). The maximal binding (B_max_) of CS_L_ to NT were 0.59, 0.69 and 0.65 mol/mol at 25 nM, 10 µM and 1 mM [Ca^2+^], respectively, which suggested that the interaction between CS_L_ and NT appeared to follow the one-site model of Hill’s equation (Table 2). Then, the effect of H_2_O_2_ on the binding of CS_L_ to NT was examined. The binding of CS_L_ to NT was not affected by 10^−6^–10^−1^ M H_2_O_2_ (Figure 4K,L, Table 1). All these results showed that the binding of CS_L_ to the IQ domain, but not PreIQ or NT, was affected by H_2_O_2_ treatment, suggesting that the IQ domain was involved in the regulation of OS on the interaction of CS_L_ with Ca_V_1.2.

### 2.5. CS_L_54-64 Was the Binding Region of CS_L_ to Ca_V_1.2

Molecular docking was used to investigate the binding properties of CS_L_ to Ca_V_1.2. The results of epitope analysis are shown in Table 3, and the count represented the amount of referenced docking pose clusters. The 47, 26 and 10 docked poses of the interaction of CS_L_ with PreIQ/IQ/NT were displayed and the most likely binding postures with the highest score among the five clusters were drawn in blue (Figure 5(A1,B1,C1)), and the corresponding fingerprints are shown in Figure 5(A2,B2,C2), respectively. PreIQ/IQ/NT were closely bound to CS_L_ with different positions and the binding region on CS_L_ was limited to a fixed area, which is the most obvious in Figure 5(A3,B3,C3). Furthermore, while binding to PreIQ/IQ/NT, the most likely residues of CS_L_ were all located in CS_L_54-64 (Figure 5(A5,B5,C5)). And the binding free energy of NT and CS_L_54-64 was significantly higher than that of PreIQ/IQ, indicating that the binding affinity of NT to CS_L_54-64 was the lowest. The binding free energies of amino acids Glu 1612 and Lys 1620 on PreIQ, Glu 1655 on IQ and Cys 136 on NT to CS_L_54-64 were minimal, suggesting that the amino acids were the main sites responsible for the binding of Ca_V_1.2 to CS_L_54-64 (Table 4). These results indicated that the binding region of CS_L_ to Ca_V_1.2 was located on Glu 54-Lys64.

## 3. Discussion

In this study, the effect and mechanism of Ahf-caltide, a polypeptide derived from CS_L_54-64, on myocardial OS injury were investigated. Ahf-caltide alleviated myocardial OS injury, assessed with decreased LDH content, MDA content, BNP level, ROS level and increased heart rate, coronary flow, cell survival and SOD activity. Ahf-caltide permeabilized into H9c2 cells and increased Ca_V_1.2, Ca_V_β2 and CAST levels by inhibiting protein degradation. CS_L_54-64 was predicted to be responsible for the binding of CS_L_ to Ca_V_1.2. At different [Ca^2+^], the binding of CS_L_ to CT1/IQ motifs on the α_1_ subunit of Ca_V_1.2 was increased in a H_2_O_2_ concentration-dependent manner. These results suggested that Ahf-caltide exerts a cardioprotective role in myocardial OS damage by upregulating Ca_V_1.2 channel expression.

In our study, the changes in the expression of α_1_ subunit and β_2_ subunit of Ca_V_1.2 channel were observed in three OS models. In CoCl_2_ and H_2_O_2_ models, the α_1_ subunit protein level was decreased, which was consistent with the previous studies. The density of Ca_V_1.2 channel estimated by dihydropyridine binding assay was decreased at 3 and 8 weeks after myocardial infarction, respectively [27,28]. In contrast, in the IR model, α_1_ subunit expression did not change, consistent with the report of Zucchi R et al. [29]. In addition, β subunit is an important regulator of Ca_V_1.2 channel, with the function of enhancing the channel surface expression by inhibiting α_1_ subunit degradation and supporting voltage gating [30,31]. In the CoCl_2_ model, β subunit expression was decreased, which was in line with the previous report that β subunit expression was downregulated in patients with atrial fibrillation [30]. Interestingly, in H_2_O_2_ models, unlike the decrease in Ca_V_1.2 level, Ca_V_β2 level was not altered. Cell death has been demonstrated to be a fundamental process in cardiovascular diseases [2,4]. However, the different changes in α_1_ and β subunit expression might not be simply attributed to the loss of cardiomyocytes that occurred in ischemia and reperfusion.

During OS, Ca^2+^ influx into cardiomyocytes mediated by the Ca_V_1.2 calcium channel results in Ca^2+^ overload and eventually leads to cell injury [6,32]. Current therapeutic strategies for myocardial IR injury focus on the inhibition of Ca_V_1.2 channel activity. However, it cannot be ignored that clinically, heart failure with reduced ejection fraction and atrial fibrillation commonly coexist, and most calcium channel blockers are not recommended in heart failure with reduced ejection fraction [33,34]. Inhibition of the Ca_V_1.2 channel activity would cause a shortening of action potential duration and refractoriness, exacerbating the development of atrial fibrillation [35]. In addition, the negative inotropic effect of Ca_V_1.2 channel blockers results in impaired systolic function, which limits its use in the clinic [34,36,37]. The negative inotropic effect is mediated by downregulation of the Ca_V_1.2 channel activity [38]. Therefore, from a long-term perspective, some patients with heart failure might not benefit from calcium channel blockade [33]. The Ca_V_1.2 channel plays an important role in regulating signaling pathways, and Ca^2+^ influx through the channel triggers the expression of a group of Ca^2+^-regulated genes, which are necessary for cell survival [39,40]. Studies have shown that the activation of Ca_V_1.2 channel reduced neuronal cell death after ischemia and decreased OS in cortical neurons [41,42]. Therefore, we believed that opening the Ca_V_1.2 calcium channel, rather than blocking the channel, may be beneficial in alleviating myocardial OS injury. In the present study, Ahf-caltide exerted a myocardial protective role by upregulating the expression of the Ca_V_1.2 channel α_1_ and β subunits. To our knowledge, this is the first report that states that the upregulation of the Ca_V_1.2 channel could alleviate myocardial OS injury.

In addition, the mechanism by which Ahf-caltide increased Ca_V_1.2 channel expression was studied, and it was shown that Ahf-caltide penetrated into H9c2 cells and was located in the cytoplasm. Ahf-caltide with amino acid residues, 3Glu, 1Gl, 3Ly, 2Pro, 1His and 1Thr exhibited strong hydrophilicity, which promotes phospholipid scrambling and membrane leakage [43]. Moreover, Lys and Glu are involved in the generation of locally polar regions inside the bilayers, which reduces the activation energy for flip-flop, therefore altering phospholipids passing through the hydrophobic passage [44]. In the presence of protein synthesis inhibitors, Ca_V_1.2 and Ca_V_β2 levels were increased by Ahf-caltide, indicating that Ahf-caltide inhibited protein degradation. Intracellular protein degradation occurs via three mechanisms, including the ubiquitin–proteasome system, lysosomal and autophagy pathway and calpain system [24]. In the present study, Ca_V_1.2 and Ca_V_β2 levels were increased in the presence of MG-132, rather than leupeptin, indicating that the proteasome pathway was involved in the degradation of the channel, which is consistent with the previous report [45]. Ca_V_1.2 and Ca_V_β2 levels were further increased by Ahf-caltide in the presence of MG-132, suggesting that the upregulation of Ca_V_1.2 channel expression by Ahf-caltide was due to its inhibition on protein degradation mediated by proteasome (Figure 6).

The calpain system consists of a family of calpain proteases and the endogenous calpain inhibitor CAST [24]. Studies have shown that in IR injury, after CAST activity was inhibited, calpain was activated and the Ca_V_1.2 channel protein was cleaved, resulting in an irreversible run-down of the channel [46]. An increase in the intracellular Ca^2+^ and calpain activity also result in CAST proteolysis [47]. In this respect, CAST is not only a downstream effector of the Ca_V_1.2 channel, but also an upstream regulator of the Ca_V_1.2 channel. Therefore, CAST expression was also examined in this study. Consistent with previous studies, in IR and CoCl_2_ models, CAST level was decreased [47,48]. However, in the H_2_O_2_ model, CAST level was upregulated. The difference in CAST expression among OS models might be related to the interaction and functional cross-talk between CAST and the Ca_V_1.2 channel. In addition, under MG-132 or leupeptin treatment, CAST level was upregulated, indicating that both the lysosomal pathway and proteasome pathway were involved in the proteolysis of CAST, which was inhibited after Ahf-caltide treatment. Therefore, Ahf-caltide might increase the CAST level by inhibiting the lysosomal pathway and proteasome pathway, thereby elevating the Ca_V_1.2 level (Figure 6).

It was reported that the CT of α_1_ subunit was sensitive to proteolysis [49]. Ahf-caltide was designed to target the α_1_ subunit, so the interaction between Ahf-caltide and the α_1_ subunit was explored during OS. Considering that Ahf-caltide was small and difficult to detect, the binding of CS_L_ to the α_1_ subunit was examined instead. The pathological level of H_2_O_2_ (>10^−7^ M) leads to OS [50,51]. With the increase in [Ca^2+^] from 25 nM (low [Ca^2+^] level) to 10 μM (high [Ca^2+^] level) and then to 1 mM (a saturating [Ca^2+^] level), the binding of CS_L_ to CT1 was enhanced by H_2_O_2_ (10^−6^–10^−2^ M) [50,51,52,53,54,55]. These results suggested that the binding of CS_L_ to CT1 was enhanced with the increase in OS at pathological [Ca^2+^]. In addition, under the 10^−1^ M H_2_O_2_ treatment, the binding of CS_L_ to CT1 was slightly decreased and the position of the bound CS_L_ significantly shifted up, which might be due to the structure of CS_L_ altered by the high concentration of H_2_O_2_. Moreover, IQ, not PreIQ or NT, was involved in the effect of H_2_O_2_ on the binding of CS_L_ to Ca_V_1.2.

In this study, CS_L_ could bind to CT1/PreIQ/IQ and the binding of CS_L_ to CT1 was in a [Ca^2+^] dependent manner, which were consistent with our previous findings [15]. NT regulates the inactivation of the Ca_V_1.2 channel [26,56]. CS_L_ was found to bind to NT, suggesting that NT might be one of the targets of CS_L_ in the regulation of the Ca_V_1.2 channel activity. Next, molecular docking was used to predict the binding properties of CS_L_ to Ca_V_1.2, and the binding region was found to be mainly located in CS_L_54-64. The binding affinity of CS_L_54-64 to NT was significantly lower than that to CT, suggesting that CT was the main target for CS_L_ in the regulation of the Ca_V_1.2 channel activity. In addition, amino acids Glu 1612 on PreIQ and Glu 1655 on IQ were found to be the main binding sites of CS_L_ to Ca_V_1.2. In the previous studies, both CS_L_ and CaM were found to bind to CT1 [15,16]. CS_L_ suppressed CaM binding to IQ and did not directly interact with CaM, suggesting that CS_L_ and CaM might share the binding sites on the CT of Ca_V_1.2 [16,27]. Glu 1612 on PreIQ and Glu 1655 on IQ are involved in the binding of CaM to Ca_V_1.2 [24,57]. The predicted sites on Ca_V_1.2 of CS_L_ in our study were consistent with the previous studies [15,16,25]. Combined with pull-down assay results, the inhibition of Ca_V_1.2 proteolysis mediated by Ahf-caltide during OS was most likely due to the interaction between the peptide and CT.

## 4. Materials and Methods

### 4.1. Materials

Ahf-caltide (amino acid sequence: EGKPKEHTEPK) and its scramble peptide were synthesized by NJ Peptide (Nanjing, China). H9c2 cells were purchased from the National Collection of Authenticated Cell Cultures (Shanghai, China). Hydrogen peroxide (H_2_O_2_) was purchased from Suicheng Pharmaceutical Co., Ltd. (Zhengzhou, China, H41024161). MG-132 (#133407-82-06), leupeptin (#103476-89-7),Tween 20 (T8220) and Hoechst 33342 (C0030) solution were purchased from Solarbio (Beijing, China), and cobalt chloride (CoCl_2_) was purchased from LEAGENE (Beijing, China, #7791-13-1). DMEM medium was purchased from Hyclone (South Logan, UT, USA, XB01). Fetal bovine serum was purchased from SERA/PRO (Immenhausen, Germany, S601S-500). CCK-8 assay kit was purchased from Biosharp (Hefei, China, BS350A), and lactate dehydrogenase (LDH, A0210-1), malondialdehyde (MDA, A003-1-2) and superoxide dismutase (SOD, A001-3-2) detection kits were purchased from Nanjing Jiancheng Bioengineering Institute (Nanjing, China). ROS detection kit (S0033S), RIPA lysis buffer (P0013B) and Coomassie brilliant blue R (ST031) were purchased from Beyotime Biotechnology (Shanghai, China). PVDF membrane was purchased from Immobilon (Darmstadt, Germany, NO. IPVH00010). Anti-BNP antibody was purchased from Proteintech (Shanghai, China, Ag28102), anti-CAST antibody from Cell Signaling Technology (Danvers, MA, USA, #4146), anti-Ca_V_1.2 antibody from Alomone labs (Jerusalem, Israel, ACC-003), anti-Ca_V_β2 antibody from Abcam (Cambridge, UK, ab 139528) and anti-GAPDH antibody was purchased from TransGen Biotech (Beijing, China, HC301). Glutathione Sepharose 4B (#17-0756-01) and PreScission protease (#169452310) were purchased from GE Healthcare (Pittsburgh, PA, USA). Protease inhibitor cocktail was purchased from Roche (Basel, Switzerland, #11697498001).

### 4.2. Animal Model Preparation

Healthy male sprague-Dawley rats, weighing 200–250 g, were purchased from Beijing Huafukang Biotechnology Co., Ltd. (Beijing, China). and the production license number was SCXK (Jing) 2019-0008. The rats were raised in the Experimental Animal Center of China Medical University and housed for three days under standard laboratory conditions. All experimental procedures were conducted in accordance with the Animal Protection and Use Committee of China Medical University and the application approval number was CMU2020175.

To investigate the effects of Ahf-caltide on hearts subjected to H_2_O_2_, the rats were randomly divided into three groups: control group, H_2_O_2_ group and Ahf-caltide group, with six rats in each group. The rats were anaesthetized by intraperitoneal injection of 1% pentobarbital sodium (40 mg/kg). Then, the hearts were mounted on a Langendorff apparatus and perfused at a constant flow (10 mL/min) with Krebs–Henseleit (K-H) buffer containing (in mmol/L) 11.10 glucose, 118.50 NaCl, 25.00 NaHCO_3_, 4.70 KCl, 1.20 KH_2_PO_4_, 2.50 CaCl_2_ and 1.20 MgSO_4_. The perfusate was gassed with 95% O_2_ and 5% CO_2_ (pH 7.4). In the control group, the hearts were perfused with K-H buffer for 60 min. In the H_2_O_2_ group, the hearts were perfused with K-H buffer for 15 min, followed by 200 μM H_2_O_2_ for 60 min. In the Ahf-caltide group, the hearts were perfused with 200 μM H_2_O_2_ and 10 μM Ahf-caltide for 60 min (Figure 1A).

To further investigate the effects of Ahf-caltide on hearts subjected to IR, the rats were randomly divided into five groups: control group, IR group and high-dose, medium-dose and low-dose of Ahf-caltide groups. In the control group, the hearts were perfused with K-H buffer purely. In the IR group, the hearts were perfused with K-H buffer for 15 min, followed by 45 min of no-flow ischemia and 60 min of reperfusion. In the peptide treatment groups, the hearts were perfused with Ahf-caltide (0.3, 1 or 10 μM) for 10 min before ischemia and during reperfusion (Figure 2A).

### 4.3. Cell Culture and OS Model Preparation

H9c2 cells were cultured in the DMEM medium supplemented with 10% fetal bovine serum and 1% streptomycin/penicillin in a humidified incubator at 37 ℃ with 5% CO_2_. The cells were randomly divided into six groups: control group, CoCl_2_ group (600 μM, 24 h) and Ahf-caltide (0.3, 1, 3 or 10 μM) groups. In the Ahf-caltide groups, cells were pretreated with Ahf-caltide for 0.5 h, followed by the exposure to 600 μM CoCl_2_ for 24 h.

### 4.4. Immunofluorescence

The immunofluorescence assay was carried out according to the instruction. Briefly, when the cells were approximately 80–90% confluent, H9c2 cells were fixed with 1% formaldehyde at 37 °C for 15 min. Then, the cells were treated with 10 μM Ahf-caltide conjugated with Rodamine (GL Biochem Ltd., Shanghai, China) and incubated for 24 h. Then, Hoechst 33342 solution was added to each well for cell staining and the cells were incubated for 20–30 min. Then, the cells were washed with PBS 2-3 times. Finally, fluorescence microscopy (Olympus, Tokyo, Japan) was used for immunofluorescence-labeled sections.

### 4.5. Cell Viability Assay

Cell viability was determined using a CCK-8 detection kit. In brief, in the 96-well plate, H9c2 cells at a density of 1 × 10^4^ cells/well were incubated with Ahf-caltide (0.3, 1, 3 or 10 μM) or 600 μM CoCl_2_. Then, 10 μL of CCK-8 solution was added to each well and the cells were incubated for 1 h. The absorbance was measured with a microplate reader at 450 nm.

### 4.6. Detection of LDH, MDA and SOD

In the 24-well plate, H9c2 cells at a density of 1 × 10^5^ cells/well were incubated with Ahf-caltide or CoCl_2_. Myocardial cell damage was assessed by detecting the LDH released into the culture medium or coronary effluent according to the instructions of the kit. The MDA content and SOD activity were detected at the end of perfusion time to evaluate the OS damage following the manufacturer’s instructions, respectively.

### 4.7. Determination of ROS

H9c2 cells (2 × 10^5^/well) were separately seeded into 35 mm cell culture dishes. After treatment with Ahf-caltide or CoCl_2_, the cells were cultured with serum-free medium and incubated with 10 μM DCFH-DA diluted using serum-free and phenol red-free DMEM at 37 ℃ for 20 min. Then the cells were washed three times with serum-free and phenol red-free DMEM. The membrane-permeable and non-fluorescent 2′,7′-dichlorofluorescin diacetate (DCFH-DA) was catalyzed by intracellular esterase and then oxidized into fluorescent DCF, which was detected by a confocal laser scanning microscope (Nikon, Shanghai, China) at an excitation wavelength of 488 nm and emission wavelength of 525 nm.

### 4.8. Western Blot Assay

After the end of experiments, the total proteins of the left ventricle or cultured cells were extracted using RIPA lysis buffer containing the protease inhibitor cocktail. The proteins were separated using SDS-PAGE and transferred onto a PVDF membrane. The membranes were blocked with 5% non-fat milk for 2 h in Tris-buffer saline containing 0.1% Tween 20 at room temperature and then were incubated with anti-BNP antibody (1:2000), anti-CAST antibody (1:1200), anti-Ca_V_1.2 antibody (1:500), anti-Ca_V_β2 antibody (1:1000) and anti-GAPDH antibody (1:1500) at 4 ℃ overnight, respectively. After washing with Tris-buffer saline containing 0.1% Tween 20 three times, the membranes were incubated with the corresponding secondary antibody at room temperature for 2 h. The blots were detected using an enhanced chemiluminescence detection solution, and then, protein abundance was quantified by chemiluminescence system.

### 4.9. GST Pull-Down Assay

The cDNA corresponding to NT (a.a. 6–140), CT1 (a.a. 1509–1789, proximal portion of CT), PreIQ (a.a. 1599–1639) and IQ (a.a. 1648–1668) motifs of Ca_V_1.2 (GenBankTM AB016287) and CS_L_ (a.a. 3–148, GenBankTMD16217) were inserted into the pGEX6p-3 vector and expressed as glutathione S-transferase (GST)-fusion proteins in Escherichia coli BL21 [15,58]. The proteins were purified by Glutathione Sepharose 4B beads and the GST region of the CS_L_ protein was removed using PreScission protease. To detect the interaction between NT and CS_L_, GST-NT (15 μg) was incubated with CS_L_ (in μM: 0, 0.1, 0.3, 1.0, 3.0, 5.0, 10.0, 30.0) for 4 h in 400 μL Tris buffer at different [Ca^2+^] (25 nM, 10 μM, 1mM). To detect the binding characteristics of CS_L_ to Ca_V_1.2 during OS, GST-NT/CT1/PreIQ/IQ (15 μg) was incubated with 10 μM CS_L_ in 400 μL Tris buffer at 4 °C for 4 h with agitation in the presence of H_2_O_2_ (in mol/L: 0, 10^−6^, 10^−5^, 10^−4^, 10^−3^, 10^−2^, 10^−1^). Bound CS_L_ to each protein was resuspended in SDS sample buffer at 100 °C for 5 min and then was separated on SDS–polyacrylamide gel. Proteins were visualized by Coomassie brilliant blue R staining and quantified by Image J software V1.8.0.112 (National Institutes of Health, Bethesda, MD, USA).

The binding curve of CS_L_ to GST-NT was fitted with the one-site model of Hill’s equation in SigmaPlot 10.0 software. The bound CS_L_ (y) was expressed as follows: y = B_max_·X/(K_d_ + X), where B_max_ is the maximum binding, X is the concentration of the free ligand and K_d_ is the apparent dissociation constant.

### 4.10. Molecular Docking

The three-dimensional structures of **CS_L_** (PDB:3DF0), PreIQ (PDB: 3G43) and IQ (PDB: 2BE6) were downloaded from the Protein Data Bank. The coding sequences of NT were obtained from the National Center for Biotechnology Information, and then, I-TASSER homology modeling server (https://zhanglab.ccmb.med.umich.edu/I-TASSER/, (accessed on 20 March 2021) was used to model the structure of the NT protein [59].

The binding of **CS_L_** or CS**_L_**54-64 to NT/PreIQ/IQ was simulated by the protein–protein docking tool of the Molecular Operating Environment (MOE, Chemical Computing Group Inc., Montreal, QC, Canada). Epitope analysis and protein–ligand interaction fingerprints analysis were used to calculate the active binding sites of **CS_L_** to NT/PreIQ/IQ. The fingerprint was built on the interactions between **CS_L_** and NT/PreIQ/IQ, such as hydrogen bonds, ionic interactions and surface contacts and so on, as shown in Figure 5(A2,A3,B2,B3,C2,C3) [11]. The binding sites with the highest binding score were chosen based on the lowest binding free energy of the protein–protein complex.

### 4.11. Statistical Analysis

Data were expressed as the mean ± standard deviation. The Student’s *t*-test was used for comparisons between two groups, and one-way ANOVA with the Dunnett’s test was used for multiple data comparisons. *p* < 0.05 was considered statistically significant.

## 5. Conclusions

In conclusion, Ahf-caltide has a protective effect against myocardial injury induced by OS. Ahf-caltide increased the Ca_V_1.2 channel α_1_ subunit, β subunit and CAST levels by inhibiting protein degradation during OS. CS_L_54-64 was responsible for the binding of CS_L_ to the Ca_V_1.2 channel IQ motif. In addition, with the increase in OS, the binding affinity of CS_L_ to IQ, rather than PreIQ or NT, also increased. Therefore, Ahf-caltide is a promising therapeutic polypeptide to restore calcium homeostasis, and the present study provides a new direction and strategy for the treatment of myocardial OS injury. This perspective of increasing Ca_V_1.2 expression is clinically important, since it offers a new direction to treat patients with heart failure, who were discharged on a contraindicated calcium channel blocker.

## Figures and Tables

**Figure 1 ijms-24-15729-f001:**
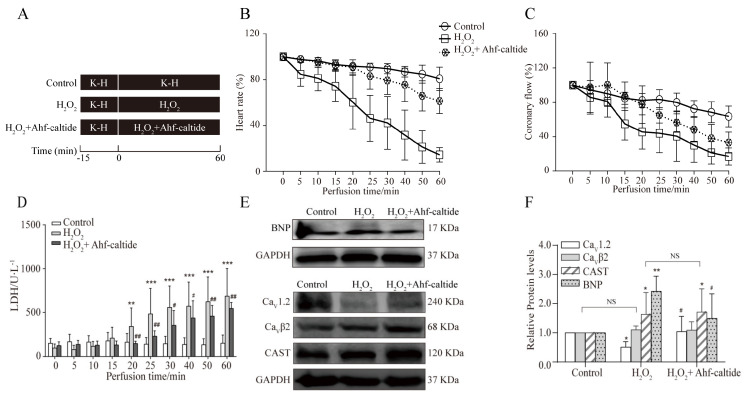
The protection of Ahf-caltide on the Langerdoff-perfused rat heart subjected to H_2_O_2_ treatment. (**A**) Schematic diagram of the experimental protocol. (**B**,**C**) Time course of changes in heart rate (**B**) and coronary flow (**C**) among groups. (**D**) LDH activity of coronary effluent in groups. (**E**,**F**) Representatives immunoblots (**E**) and quantitative analysis (**F**) of Ca_V_1.2, Ca_V_β2, CAST and BNP levels in hearts of groups. * *p* < 0.05, ** *p* < 0.01 and *** *p* < 0.001 vs. control group; # *p* < 0.05 and ## *p* < 0.01 vs. H_2_O_2_ group. NS, not significant. Results represent at least five independent experiments.

**Figure 2 ijms-24-15729-f002:**
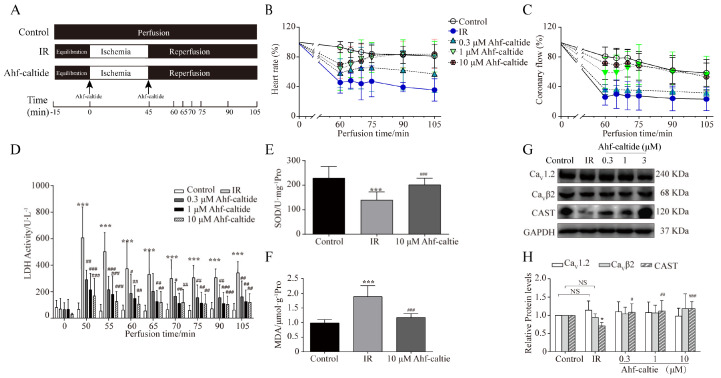
The protection of Ahf-caltide on the Langerdoff-perfused rat hearts subjected to IR injury. (**A**) Schematic diagram of the experimental protocol. (**B**,**C**) Time course of changes in heart rate (**B**) and coronary flow (**C**) among groups. (**D**) LDH activity of coronary effluent in groups. (**E**,**F**) The SOD activity (**E**) and MDA content (**F**) in the heart tissues. (**G**,**H**) Representative images (**G**) and quantitative analysis (**H**) of Ca_V_1.2, Ca_V_β2 and CAST in hearts among groups. * *p* < 0.05 and *** *p* < 0.001 vs. control group; # *p* < 0.05, ## *p* < 0.01 and ### *p* < 0.001 vs. IR group. NS, not significant. Results represent at least five independent experiments.

**Figure 3 ijms-24-15729-f003:**
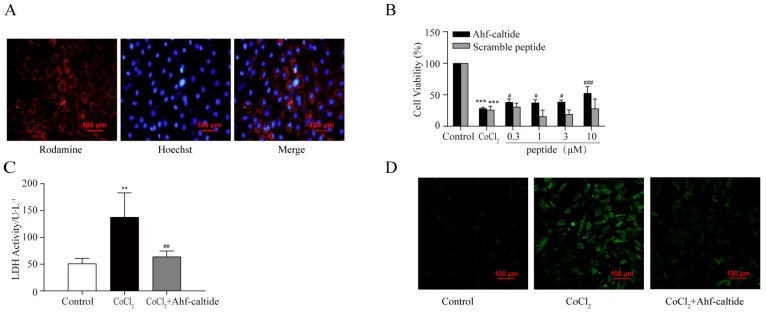
The protection by Ahf-caltide on H9c2 cell injury induced by CoCl_2_ treatment. (**A**) The localization of Ahf-caltide in H9c2 cells examined by fluorescence microscopy (scale bars, 100 μm). (**B**) The cell viability was detected by CCK-8. (**C**) LDH activity of supernatant in groups. (**D**,**E**) ROS level in groups (scale bars, 100 μm). (**F**,**G**) Representative immunoblots (**F**) and quantitative analysis (**G**) of expression levels of Ca_V_1.2, Ca_V_β2, CAST in groups. (**H**,**I**) Representative immunoblots (**H**) and quantitative analysis (**I**) of the effects of cycloheximide (CHX) on the expression of Ca_V_1.2, Ca_V_β2, CAST in groups. (**J**,**K**) Representative immunoblots (**J**) and quantitative analysis (**K**) of the effects of MG-132/Leupeptin on the levels of Ca_V_1.2, Ca_V_β2, CAST of groups. * *p* < 0.05, ** *p* < 0.01 and *** *p* < 0.001 vs. control group or as indicated on the figure; # *p* < 0.05, ## *p* < 0.01 and ### *p* < 0.001 vs. CoCl_2_ group. NS, no significance. Results represent at least three independent experiments.

**Figure 4 ijms-24-15729-f004:**
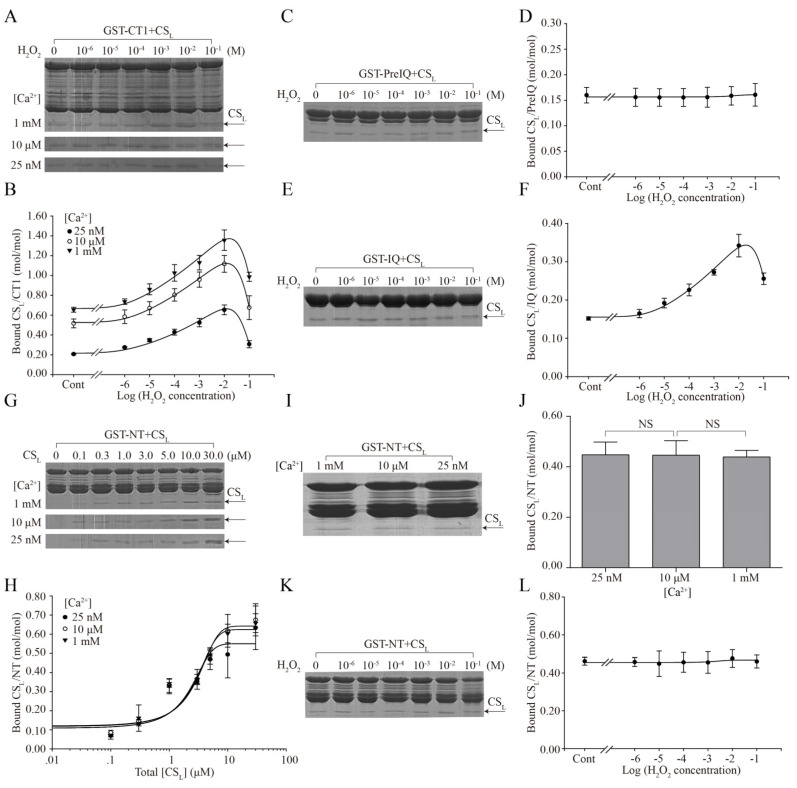
Concentration-dependent effect of H_2_O_2_ on CS_L_ binding to N- and C-terminus of Ca_V_1.2 channel α_1_ subunit. GST pull-down assay for the effect of H_2_O_2_ (10^−1^–10^−6^ M) on the binding of CS_L_ to CT1 (**A**) or PreIQ (**C**) or IQ (**E**) or NT (**K**). Plots of the effect of H_2_O_2_ with fitted curves on the CS_L_ binding to CT1 (**B**) or PreIQ (**D**) or IQ (**F**) or NT (**L**). Pull-down assay (**G**) and binding curves (**H**) for CS_L_ binding to NT. Pull-down assay for the effect of [Ca^2+^] (**I**) and quantitative analysis (**J**) on the binding of CS_L_ to NT. Symbols are as indicated in each graph. NS, no significance. Protein bands were visualized using Coomassie brilliant blue staining and indicated by arrows. Results represent at least four independent experiments.

**Figure 5 ijms-24-15729-f005:**
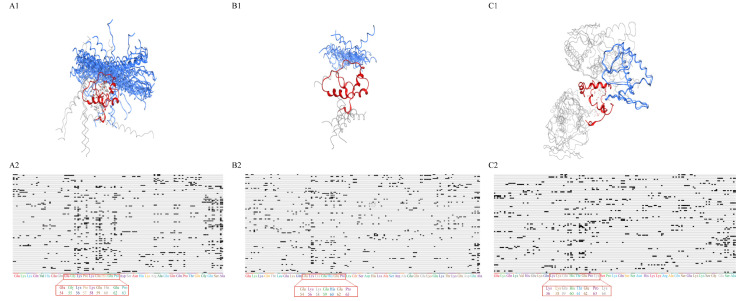
Molecular docking analysis of **CS_L_** or CS**_L_**54-64 binding to N- and C- terminus of Ca**_V_**1.2 channel α_1_ subunit. (**A**–**C**) Improved five groups of epitope clusters in Table 3 corresponding to the totally docked poses of (**A1**) CS**_L_**/PreIQ, (**B1**) CS**_L_** /IQ, (**C1**) CS**_L_** /NT; the fingerprints for the interaction obtained from the five groups’ docked poses between CS**_L_** and (**A2**) PreIQ, (**B2**) IQ, (**C2**) NT; the fingerprints for the interaction from the first group pose between CS**_L_** and (**A3**) PreIQ, (**B3**) IQ, (**C3**) NT; the docked poses of CS**_L_**54-64 binding to (**A4**) PreIQ, (**B4**) IQ, (**C4**) NT; the interaction diagram of Lys 64, Lys 56 and Glu 54 on CS**_L_**54-64 with PreIQ (**A5**), IQ (**B5**) and NT (**C5**), respectively. The diagrams in the first group with the highest score are represented by blue “spaghetti” and the others with a translucent “spaghetti”. Blue structures represent PreIQ/IQ/NT, red structures represent **CS_L_** or CS**_L_**54-64.

**Figure 6 ijms-24-15729-f006:**
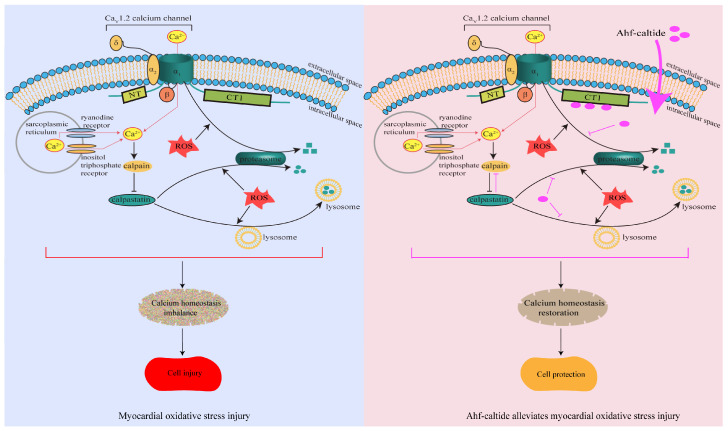
Mechanism scheme for the protection of Ahf-caltide on myocardial OS injury. The left panel is the recapitulative scheme for myocardial OS injury. The right panel is the recapitulative scheme for the effect of Ahf-caltide during OS.

**Table 1 ijms-24-15729-t001:** The effect of H_2_O_2_ on the binding of CS_L_ to GST-CT1/PreIQ/IQ/NT.

	GST-CT1 + CS_L_			GST-PreIQ + CS_L_	GST-IQ + CS_L_	GST-NT + CS_L_
[Ca^2+^]	25 nM	10 μM	1 mM	10 μM	10 μM	10 μM
[H_2_O_2_]						
0 M	0.21 ± 0.01	0.52 ± 0.02	0.65 ± 0.25	0.16 ± 0.01	0.15 ± 0.02	0.46 ± 0.01
10^−2^ M	0.65 ± 0.02	1.12 ± 0.03	1.34 ± 0.10	0.16 ± 0.01	0.34 ± 0.01	0.46 ± 0.01
10^−1^ M	0.31 ± 0.01	0.68 ± 0.05	0.99 ± 0.05	0.16 ± 0.01	0.26 ± 0.05	0.46 ± 0.01

**Table 2 ijms-24-15729-t002:** Parameters for binding of CS_L_ to GST-NT.

	GST-NT + CS_L_		
[Ca^2+^]	25 nM	10 μM	1 mM
B_max_, mol/mol	0.59	0.68	0.65
*K_d_*, μM	1.11	1.48	1.32
*R* ^2^	0.98	0.98	0.98

*K_d_*, apparent dissociation constant, B_max_, maximum binding, *R*^2^, coefficient of determination.

**Table 3 ijms-24-15729-t003:** Epitope Analysis for CS_L_ as a receptor binding to PreIQ/IQ/NT.

	Count	Score	Residue List
CS_L_ + PreIQ			
1	41	−7.85	K56 K58 H60 E62 P63 E112 S113 A115
2	2	−6.25	E36 K37 E54 G55 S87 A88 Q91 P92
3	1	−5.70	E59 K68 Q69 D72 N76 H79
4	2	−5.65	Q53 E54 K56 A88 E89 Q91 P92
5	1	−5.49	E36 Q40 T42 S85 T94 K95 D105
CS_L_ + IQ			
1	20	−7.17	K56 P57 K58 H60 E62 P63 A115
2	1	−6.14	K68 Q69 N76 H79 A83 E94 K95
3	1	−5.91	E36 S39 Q40 K43 K51 E98 T91
4	3	−5.55	Q40 T42 L44 K100 T101 K102 D105
5	1	−5.41	S39 T42 K43 E98 K100 T101 K102
CS_L_ + NT			
1	2	−6.46	E62 P63 K64 S65 K68 E112 A115
2	4	−6.29	E36 K37 S39 Q40 K83 S85 K102
3	1	−6.10	G55 K56 K58 E59 Q89 S93
4	1	−6.05	E59 E62 D72 T73 N76 N90 K95
5	2	−5.96	T61 K68 Q69 N76 H79 S93 E94

**Table 4 ijms-24-15729-t004:** Predicted binding sites of CS_L_54-65 to Ca_V_1.2 channels.

Groups	CS_L_54-65	Ca_V_1.2 Channel	Interaction Types	E (kcal/mol)
CS_L_54-65 + PreIQ	Lys 64	Lys 1620	Ionic	−4.70
	Pro 63	Arg 1624	H-acceptor	−3.60
	Lys 58	Glu 1612	H-donor/Ionic	−2.40
	Gly 55	Glu 1612	H-donor	−2.30
CS_L_54-65 + IQ	Lys 56	Glu 1655	Ionic	−8.10
	Thr 61	Arg 1663	H-acceptor	−3.40
	Lys 64	Lys 1661	H-donor	−0.50
CS_L_54-65 + NT	Glu 54	Cys 136	H-donor	−1.80
	Glu 54	Thr 138	H-acceptor	−0.90
	Lys 64	Ser 123	H-donor	−0.70

## Data Availability

All data generated or analyzed during this study are included in this published article.

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
