# Peer review of "Ahf-Caltide, a Novel Polypeptide Derived from Calpastatin, Protects against Oxidative Stress Injury by Stabilizing the Expression of CaV1.2 Calcium Channel"

_ijms, 2023, doi:10.3390/ijms242115729_

Round 1

Reviewer 1 Report

Comments and Suggestions for Authors

The study IJMS- 2664505 by Xue et al. investigated the potential protective effects of Ahf-altitude against myocardial injury-induced oxidative stress. The study found that Ahf-altitude restored calcium homeostasis by targeting the CaV1.2 calcium channel and its regulating factor, calpastatin (CAST). As a result, lactate dehydrogenase, malondialdehyde, and reactive oxygen species decreased, while heart rate, coronary flow, cell survival, and superoxide dismutase activity during oxidative stress increased. Additionally, Ahf-altitude increased CaV1.2, CaVβ2, and CAST levels by inhibiting protein degradation. This interesting study suggests this could be a promising new treatment option for myocardial oxidative stress injury. My suggestions for revisions are as follows:

  1. The introduction section in paragraph 1 should include more detail and cite related literature.
  2. Please label all Figures that need to be marked clearly and increase the font size and high-resolution Figures, particularly Figure 5.
  3. Check for subscripts and superscripts throughout the manuscript; for example, [Ca2+], 2+ should be in superscript.
  4. Please briefly describe this study's translational and clinical aspects in the conclusion. 
  5. Also, check the manuscript for grammar and spelling.

Comments on the Quality of English Language

Please check the manuscript for grammar and spelling.

Reviewer 2 Report

Comments and Suggestions for Authors

Authors present the effects of a peptide as a target of calcium channels in ischemia-reperfusion injury. While the paper is well written, I have major and minor comments below that are required to address prior to being appropriate for publication.

Major comments:

Regarding anti-CaV1.2: I have experience with this antibody from Alomone labs and know from experience that it is highly non-specific. Authors need to validate that they are indeed detecting the protein of interest with a positive and negative control lysate. Considering the quality of the bands for this protein, my concerns are further warranted.

Regarding Figure 1E: BNP and GAPDH are extremely poor. After assessing the original blots, it appears that proteins were poorly transferred to your membrane considering the pattern of inconsistency of chemiluminescence and weak GAPDH expression. This is not acceptable or reliable. If must be repeated.

Figure 3D: Nuclear counterstain for ROS images are required.

Minor Comments:

Authors refer to H9c2 cells as cardiomyocytes throughout the text. H9C2 cells are not cardiomyocytes since you have not differentiated them, they are merely myoblasts. In fact, if confluent, they become skeletal muscle cells.

Line 434: in what buffer?

For Ahf-caltide synthesis in methods: Can you provide the amino acid sequence?

Section 4.1: Please provide catalog numbers for all kits, compounds, and antibodies.

Section 4.2: What was the sex of these animals?

Section 4.3: Seeding densities and plate sizes must be indicated.

Line 412: What was the concentration of CoCl2? authors should provide HIF-1 alpha protein data to validate the model is working in their lab.

Section 4.4: The details of cell fixing and staining should be described in detail.

Section 4.5: Details of what compounds were assessed with cell viability should be provided here.

It is not clear which part of the heart was used for protein analysis.

Were any inhibitors added to lysis buffer?

Data should be displayed as standard deviation, not standard error. Population corrections have no value in this context and SD allows for better visualization of data spread/

Round 2

Reviewer 2 Report

Comments and Suggestions for Authors

I request that authors include the cav1.2 antibody validation in the supplement. Otherwise, I have no further comments. Authors have addressed them. However, nuclear counterstain should be considered in future works to validate cell density, as this is the primary purpose.

Round 3

Reviewer 2 Report

Comments and Suggestions for Authors

No further comments.